# Collaborative-AI Knowledge Graph Generation: Taxonomization of IATE, the EU Terminology

Michael Wetzel[1], Stelios Mathioulakis[2], Jochen Hummel[1,2], and Alena Vasilevich[1][0000−0002−9769−1885]

[1] Coreon GmbH, Berlin, Germany
michael@coreon.com alena@coreon.com
http://www.coreon.com
[2] ESTeam AB, Gothenburg, Sweden
stelios@esteam.se jochen@esteam.se
https://www.esteam.se/

**Abstract.** Formalized knowledge is a powerful resource for AI projects, but it is usually created at great expense. Taxonomization is linking a flat set of concepts into a hierarchical knowledge graph, and in this work, we present our approach to semi-automatic generation of such concept maps, elevating a sub-domain of IATE terminology into a multilingual knowledge graph. We taxonomized a flat list of concepts within the COVID sub-domain, benchmarking two approaches to tackle this task: automatic concept map creation using an enhanced ML-powered language model and manual creation of the graph by a linguist expert. We dwell on advantages of the collaborative method, made easy by a user-friendly UI, and show how the achieved productivity rate can make taxonomization of large terminology databases economically viable.

**Keywords:** Collaborative taxonomization · Knowledge Graph UI · KG Construction Methods · Machine Learning

## 1 Introduction

In the realm of data-driven businesses, structured data is a valuable resource. IATE (Interactive Terminology for Europe)[1], with almost one million concepts storing multilingual terms and metadata, holds a large part of the textual knowledge of the EU (European Union). However, it can only be accessed lexically, and the database concepts stand alone.

If IATE were taxonomized, i.e. related concepts linked up into knowledge graphs yielding a full-fledged ontology, its data could not only be consumed by linguists but would also become machine-readable, converting it into a powerful resource for AI projects, particularly within the segment of small and medium-sized enterprises (SMEs) that rarely have the means to create multilingual formalized knowledge. Building and maintenance of big taxonomies require not only vast financial support but also supervision of domain-specific experts.

---

[1] Available at https://iate.europa.eu/

The objective of this study was to investigate if, given recent scientific advances, we could facilitate a speedy creation of a deeply-structured taxonomy, turning a flat set of concepts without relations between them into a hierarchical knowledge graph. To test our hypothesis, we chose two scenarios: manual and semi-automatic taxonomization. In the first scenario, a linguist took a flat list of concepts and turned them into a structured taxonomy, establishing broader-narrower relations between the concepts. In the second scenario, an ML-powered algorithm drafted the taxonomy, which was later curated by a linguist. In both cases, the taxonomies feature not only initial input concepts but also the established relations between them and higher-level parent nodes that bring a meaningful structure into the taxonomy. Implementation

## 2    Methodology and Data

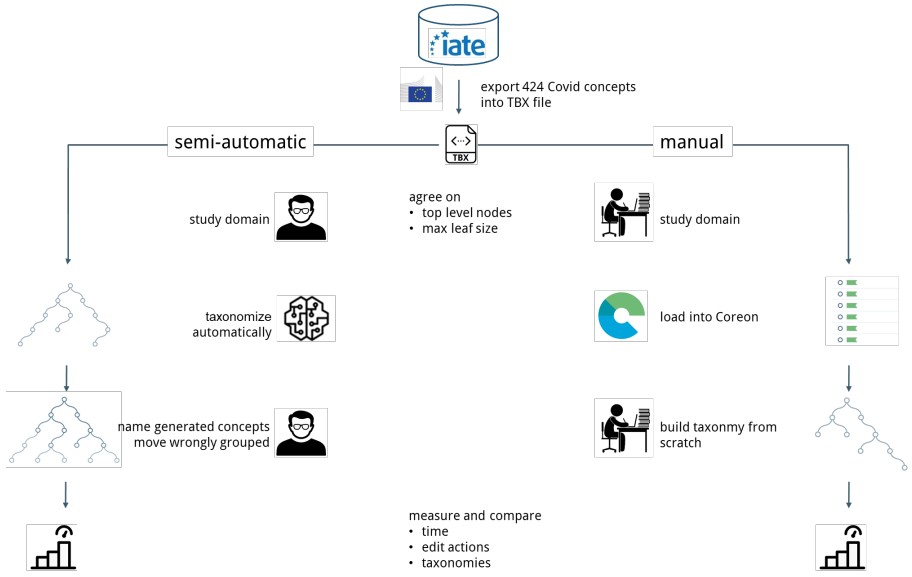

**Fig. 1.** Comparing two taxonomization approaches.

When given the same set of concepts and asked to create a taxonomy, people are likely to produce different results due to our natural diversity of views on how concepts shall be categorized. We therefore did not aim for an in-depth review of semantic coherence within the created concept clusters, focusing rather on comparison of measurable parameters, such as working hours, amount of "transactions" in the software, number of relations created. To make the results comparable to some degree, resulting taxonomies share 5 identical top-level

nodes (i.e. "entities", "instruments", "medicine", "social aspects", and "miscellaneous"), and parent nodes may not have more than 20-25 children. It should also be noted that the curators are not domain-knowledge experts, yet they were well familiar with the used software. Figure 1 illustrates the steps executed for both scenarios. Exported 424 IATE Covid concepts, used in both cases as initial input, were identical.

## 2.1 Semi-automatic Taxonomization

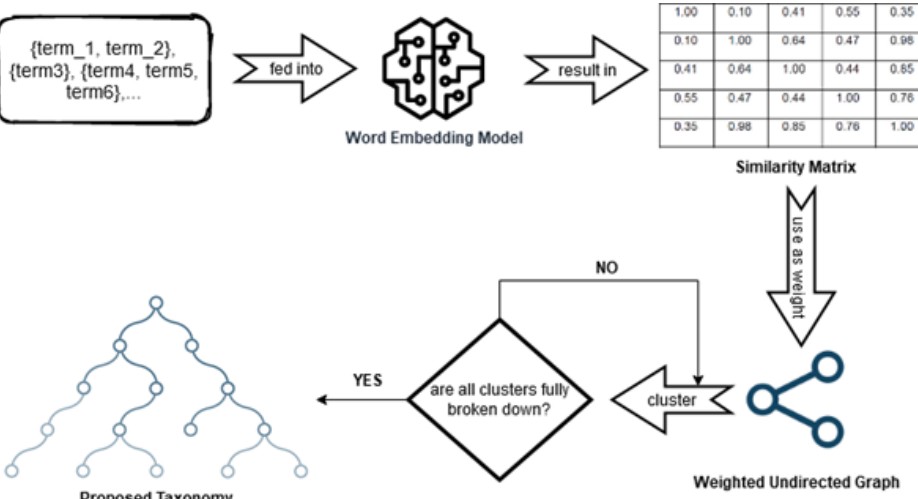

**Fig. 2.** ML-driven taxonomization process.

We have developed a novel ML algorithm for automatic taxonomy induction, using context-free tokens, i.e. terms and synonyms comprising each IATE Covid concept, as input data. First, input is vectorized, so text tokens representing each concept are converted into dense vectors of numbers. We used a pre-trained and publicly available FastText word embedding (WE) model to convert terms into vectors [3]. Next, we calculate a pairwise cosine similarity for all WE combinations, forming a symmetric matrix of similarities. To create a graph, each node is represented by an input concept, and to form the vertices/edges of the graph, a threshold is used, i.e. a median cosine similarity of all the given similarities. If two concepts have a similarity higher than the determined threshold, an edge is formed between the nodes that correspond to these two concepts. The formed edge will then have the similarity score as a weight attribute. Ultimately, all combinations of concept nodes are checked to see whether a connection between them will be formed, hence resulting in an undirected weighted graph.

Subsequently, Louvain algorithm is used on the aforementioned graph to reveal nodes that are more closely related than others [1, 2]. The process is executed recursively in order to further break down clusters into subclusters; with every such division, an intermediate node is created until there is nothing to break down. The produced taxonomy starts with a root node and expands until the leaf nodes reach the input concepts. The automatically generated intermediate branch nodes are labelled with temporary IDs that are to be named manually. The algorithm is essentially forming concept clusters, as concept nodes form clusters with a common parent node. Also, neighbouring cluster concepts are semantically closer than clusters farther apart in the taxonomy. Figure 2 summarizes the described taxonomization process; input concepts are represented by sets of terms in curly brackets.

The resulting taxonomy was revised by a curator; input concepts became leaf nodes, grouped into the generated 55 higher level nodes with unassigned temporary labels. Curator's approach was to traverse clustered leaf nodes and assign meaningful names to each of 55 parent concepts, replacing temporary labels. Table 1 quantifies the human effort spent on the curation of the automatically created taxonomy. Even though we did not quantify the semantic soundness of the generated clusters, it is worth noting that most of them were pretty accurate, whereas grouping errors were likely triggered by corpora, used to pre-train ML models we utilized for taxonomization (e.g., 'interstitial space', space between cells, and 'hospital pharmacy' were wrongly clustered together, as for WE both concepts are spaces, appearing in the similar semantic neighborhoods).

### 2.2   Manual Taxonomization

When approaching manual taxonomization, curator's strategy was to move from concept to concept, starting from the top level nodes and establishing broader-narrower relations, later dividing the nodes further down in sub-trees. This top-down method requires several passes through the data. Initially, concepts were sorted alphabetically which meant the taxonomist had to constantly jump from one context to another (e.g., from 'border worker' to 'bronchoalveolar lavage' to 'budding'– concepts that are semantically far off). Table 1 provides the effort metrics of manual taxonomization (excluding all preparatory steps and domain learning curve). While being a tedious challenge, the created taxonomy is a pure result of the intellectual human capacity, not influenced by additional obscure models or primed by pre-trained algorithms.

## 3   Results

Coreon UI[2] was used for both building and curation of the presented taxonomies. Since it records any modifications in the data repository, change logs were retrieved and analyzed to determine precisely what actions the taxonomists performed and how much time was spent. Table 1 demonstrates such information

---

[2] https://www.coreon.com/

for both scenarios. When comparing the numbers yielded by both approaches, we see that starting with a pre-drafted knowledge graph, the curator was five times faster, not only counting pure working hours but also when comparing the amount of transactions, i.e. events, mouse-clicks in the software.

**Table 1.** Taxonomization Effort.

| Metrics | Manual Taxonomization | Semi-automatic Taxonomization |
|---|---|---|
| Curator's time spent | 40h | 8h |
| Number of relations changed | 1417 | 432 |
| Number of new concepts created | 115 | 28 |

When starting with an automatically pre-drafted taxonomy, curators are not faced with a long list of to-be-processed concepts; they rather work cluster by cluster. Even when clusters are off, the topic and focus are stable, and there is no jumping between contexts (see Figure 3). Also, the algorithm encourages

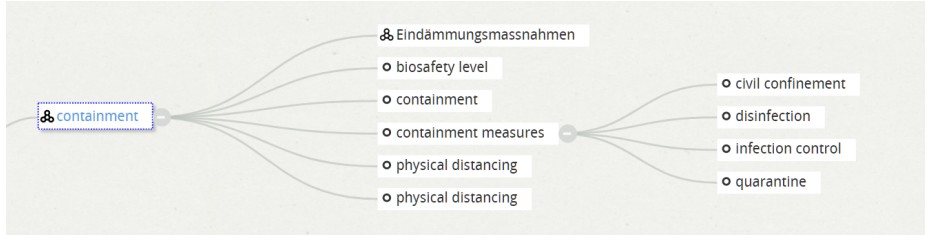

**Fig. 3.** IATE concept 'containment measures' becomes a group concept.

a strictly hierarchical tree-like concept system (mono-hierarchical in this case), whereas in the manual method, the curator often hooked concepts under more than one parent. Our collaborative-AI approach can also facilitate parallel working, allowing distributed curation of separate sub-trees. Ultimately, the curator benefits from the UI that allows to comfortably restructure the resulting knowledge graph (see Figure 4).

## 4   Conclusion

We demonstrated that the suggested collaborative taxonomization approach – combining ML and human curation – can significantly bring down the effort while

yielding a taxonomy of a consistent quality. Given availability of supporting textual data and a reasonable project size to cover for set-up expenses, the achieved performance and resource-saving advantages of our custom method makes taxonomization of even larger terminology databases economically viable. Every taxonomization project revolves around a specific domain, dictated by the vocabulary of the terms at hand (virology, general medical terms, etc.). To create more accurate vector representations (WE) of those tokens, the specific context in which they apply should be exploited. Data related to the domain at hand can be used to re-train WE model instead of using a more generic pre-trained one, primed by generic Wikipedia texts. Literature suggests that such re-training usually yields better results [4], and consistent presence of metadata in the terminology database can also aid the taxonomization process (e.g. metadata about a semantic parent of a term, free-text concept definitions).

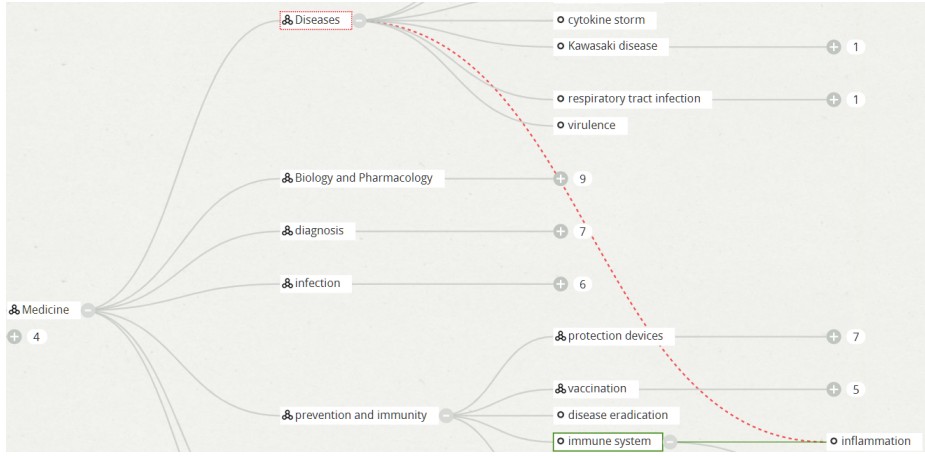

**Fig. 4.** New high-level node 'Diseases'.

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
