# OpenReview forum: "Collaborative-AI Knowledge Graph Generation: Taxonomization of IATE, the EU Terminology"
_eswc-conferences.org/ESWC/2021/Workshop/KGCW — KGCW 2021_

### Official Review · ~Sven_Lieber1 · 2021-04-13
**Clearly written paper showing the practicality of the presented approach, interesting future directions, but more distinction from related work desirable**

**Rating:** 7
**Confidence:** 3

**Review:**

This paper presents a collaborative approach to create taxonomies from flat list of hierarchies
by leveraging a machine learning algorithm generating taxonomies which are refined by an expert.
The approach was compared to a manual curation by an expert and the results indicate that an expert is five times faster using the presented approach.


## Summary


The paper provides a practical solution to an existing problem, it is well structured and easy to follow
and the presented approach involves users which fits the workshop's special focus.

The results are not too surprising comparing to a manual curation.
It would have been more interesting to see a comparison to other (semi) automatic generation solutions,
or at least some argumentation why existing approaches are not feasible in the presented use case.
Similarly, the design choices of the novel algorithm are not justified, given that there is already related work [1,2,3].
Why was this novel method invented? What are its advantages and disadvantages?

This also affects one of the conclusions: "our custom method makes taxonomization of even larger terminology databases economically viable".
Even though this is the case, other existing approaches may make it economically viable too. To which extent or in which dimension is the presented approach superior?

The last point of the conclusion is very interesting, how can existing metadata aid the taxonomization process?
How can ML methods and semantic methods be combined to achieve more correct results? How could existing resources such as EuroVoc or Wikidata be facilitated?
Future work in this direction would clearly fit this or similar venues.

Although scientifically more comparisons are needed to evaluate the novel method,
this is a short paper after all. The practicality was demonstrated,
the methodology seems sound, future directions were presented,
and the paper is well written. Thus, overall a nice contribution with interesting future perspectives.



Please find the detailed review per section below

## Introduction

The problem statement is sound: creating taxonomies is a resource intensive task and small and medium-sized enterprises (SMEs) definitely could benefit from support.
Since the introduction does not provide any references a reader is forced to believe the authors, ideally more references would back up different statements of the authors.
But the argumentation seems logical and I acknowledge the space limitation.

IATE, only a flat list of concepts, is mentioned as use case and possible candidate for taxonomization.
However, here I am missing a bit of context, the EU also provides EuroVoc (also used by IATE) which is already a taxonomy.
Does EuroVoc not provide needed information? Why is it necessary to create a taxonomy from IATE in the first place?
The tool used in the study, developed in the company of two of the authors, even provides browsing the EuroVoc taxonomy.
Providing information *why* IATE concepts need to be taxonomized would improve the paper.

## Methodology and data

Both approaches are well described. It could be better highlighted in the first paragraph of section 2 that both tasks were performed by different curators

## Results

It is not too surprising that the semi-automatic taxonomization was quicker.
It would have been interesting to compare the different depths of the solution, as the objective of the study was a "speedy creation of a deeply-structured taxonomy"

## Minor:

* I find the word "collaborative" a bit misleading for the presented approach as in fact no humans are collaborating but a human and a machine. Although there exists some human machine collaboration in robotics,
collaboration may imply that involved parties share a goal, whereas it is debatable that the presented ML approach has a goal similar to the human "collaborator"
* The introduction ends in a single word "Implementation"
* The sentence "We dwell on advantages of the collaborative method" in the abstract does not indicate which of the previously mentioned methods is meant, mentioned were only an "automatic concept map" and a "manual creation"
* phrasings like "pretty accurate" could be made more concise

[1] Iloga, Sylvain, Olivier Romain, and Maurice Tchuenté. "An efficient generic approach for automatic taxonomy generation using HMMs." Pattern Analysis and Applications 24.1 (2021): 243-262.

[2] Carrion, Belen, et al. "A taxonomy generation tool for semantic visual analysis of large corpus of documents." Multimedia Tools and Applications 78.23 (2019): 32919-32937.

[3] Sánchez, David, and Antonio Moreno. "Automatic Generation of Taxonomies from the WWW." International Conference on Practical Aspects of Knowledge Management. Springer, Berlin, Heidelberg, 2004.

---

### Official Review · ~Edna_Ruckhaus2 · 2021-04-14
**A straightforward short research paper that presents a semi-automatic ML-based method for the generation of taxonomies of concepts. A statement of the problem, a summary of the related work, and the research contribution should be included in the paper.**

**Rating:** 7
**Confidence:** 4

**Review:**

This is a short research paper that presents a Machine-Learning-based semi-automatic method to generate a taxonomy of concepts within the COVID sub-domain. It compares the proposed method with a manual taxonomy generation effort. The comparison is done in terms of the time spent by the experts and the number of relations and concepts that have been generated. The paper is related to the workshop's main topic of Knowledge Graph Construction and specifically to methods and techniques for Knowledge Graph Construction.

The paper is clearly written but lacks presentation of the problem, other approaches for its solution (a summary of the related work or as the paper states, "recent scientific advances") and what would be the contribution of this research. It says that the approach is "novel", the novelty should then be well sustained.

The semi-automatic approach (Section 2.1) based on word embeddings and the Louvain algorithm is clear and quite straightforward. The statement on the soundness of the generated clusters is vague: how do the authors determine that the results are "pretty accurate"?  with respect to what?

The work is focused on evaluating the time spent by the curators and the concepts and relations that have been created (number of transactions). However, there could have been a more in depth comparison between the concepts and relations created with the proposed approach and those created with the manual taxonomization.

The setting should be described, i.e. how many curators participated in the study . In the future, domain experts can be used in order to consider their manual taxonomization result as the Golden Truth.

There is a typo at the end of section 1: taxonomy. Implementation -> taxonomy implementation

---

### Official Review · ~Ana_Iglesias-Molina2 · 2021-04-19
**ML-based approach for semi-automatic taxonomy generation, well-written, clear and interesting**

**Rating:** 7
**Confidence:** 3

**Review:**

This paper presents an approach to generate taxonomies semi-automatically from flat lists of concepts. The purpose is facilitating the taxonomy creation process by providing an initial taxonomy, so that curators (or, in general, whoever wants to build a taxonomy) don't start from scratch. The proposed approach is evaluated with COVID concepts extracted from IATE and compared with full-manual process in terms of time spent in curation, number of relations changed and new concepts created.

I find the paper clear, well-written and easy to read, the ideas flow correctly throughout the sections. It is OK for a demo and shows promising work in the future lines discussed at the end of the paper. I did however encountered some issues. Mainly, I missed some section or paragraph showing related work. I understand the space limitations, but a brief description of the SOTA would be helpful (and welcome for non-experts in the field). I especially missed it in the election of the word embedding model, to explain the reasons on using that particular one. Regarding space issues, they may be solved optimizing the space within the figures in order to make them shorter and leaving thus more space for text.

I also missed a more detailed explanation on the experiment/evaluation setup, number of curators, and maybe more discussion about differences in the outputs from manual vs proposal (I would find that interesting at least). As the work progresses I'm sure it will be more elaborated. Overall I think this is an interesting and promising work, adequate for this workshop.

---

### Meta-Review · Program_Chairs · 2021-04-21

**Recommendation:** Accept
**Confidence:** 5

**Metareview:**

The paper was clear, well-written and easy to read by all reviewers and the ideas flow fluently. All reviewers were positive for the paper's acceptance but in the same time they all pointed that the paper misses related work. The paper would benefit from a clear discussion regarding why existing approaches are not suitable for the use case. We would thus strongly advice the authors to include some related work and a small discussion around it.

---

### Decision · Program_Chairs · 2021-04-23

Accept